# Influences of Oxygen Ion Beam on the Properties of Magnesium Fluoride Thin Film Deposited Using Electron Beam Evaporation Deposition

**Gong Zhang [1], Xiuhua Fu [1],\*, Shigeng Song [2],\*, Kai Guo [3] and Jing Zhang [1]**

[1] School of OptoElectronic Engineering, Changchun University of Science and Technology, Changchun 130000, China; zgoptics@126.com (G.Z.); zhangjing840225@cust.edu.cn (J.Z.)

[2] Institute of Thin Films, Sensors and Imaging, School of Engineering and Computing, University of the Westof Scotland, PA1 2BE Paisley, UK

[3] Optoelectronic component development, Beijing Institute of Control Engineering, Beijing 100094, China; kg_optics@126.com

\* Correspondence: fuxiuhua@cust.edu.cn (X.F.); shigeng.song@uws.ac.uk (S.S.)

**Abstract:** Magnesium fluoride ($MgF_2$) materials are commonly used for near/medium infrared anti-reflection optical coatings due to their low refractive index and wide-range transparency. Ion assistant deposition is an important technique to increase the density of $MgF_2$ and reduce absorption around 2.94 µm caused by high porosity and moisture adsorption. However, the excessive energy of Argon ion will induce a color center and; therefore, lead to UV/Visible absorption. In this paper, oxygen ion was introduced to reduce the color center effect in $MgF_2$ thin film deposited using electron beam evaporation with ion assistant. The films were deposited on Bk7 and single crystal silicon substrates under various oxygen ion beam currents. The microstructure, optical constant, film density, stress, and adhesion are investigated, including the absorption properties at the infrared hydroxyl (–OH) vibration peak. The results show that as the oxygen ion beam current increases, the absorption value at the position of the infrared OH vibration, defects, and stress of the film decrease, while the refractive index increases. However, $MgF_2$ has poor adhesion using oxygen ion-assisted deposition. Thin $MgF_2$ film without ion beam assistant was used as adhesive layer, high density, and low absorption $MgF_2$ film with good adhesion was obtained.

**Keywords:** $MgF_2$; color center absorption; density; crystal frequency; stress; adhesion

## 1. Introduction

Near/medium infrared detectors can work in both low light night vision and medium bands simultaneously, which can be widely used in military and civil fields [1–3]. The $MgF_2$ is a suitable material due to its low refractive index and transparency over the ultraviolet range to the far-infrared region [4,5]. However, the films easily absorb moisture in the atmosphere and relatively large tensile stress because of poor film structure, resulting in spectrum shift. The results show that increasing the deposition temperature of the substrate can increase the density of the film and reduce the instability caused by moisture absorption of $MgF_2$. When the deposition temperature of the substrate reaches 0.6 Tm [6] (where Tm is the melting temperature of the material), the aggregation density reaches a stable value of 0.9. Thereafter, the density will not increase with the increase of deposition temperature. Raising the temperature cannot completely solve the problem of the aggregation density [7,8]. Therefore, ion beam-assisted deposition is used to improve the density of the film. However, there are limitations to be considered in ion beam-assisted deposition. Both Matin Bischoff and Targove researches showed a technology which can produce high packing density and a low extinction coefficient by

plasma(ion)-assisted deposition, but the pure fluorine introduced in the vacuum chamber will influence the environment and also reduce the life of the vacuum chamber. M. Kennedy research showed that xenon ion bombardment has a better result than Ar ion backfill oxygen at ultraviolet–visible wavelengths, but the infrared property of films has not yet been considered in this paper [9–11]. Dumasp's research showed that the use of Ar as an ion source to assist gas deposition increased the bulk density of $MgF_2$ films, but excessive particle energy led to a change in the stoichiometric ratio of Mg and F, resulting in F atom vacancies, which led to the absorption of ultraviolet and visible parts [12,13]. These studies further show that excessive $Ar^+$ ion energy can lead to more serious moisture absorption problems when $MgF_2$ thin films are evaporated by an electron gun [14].

The working band of near/mid infrared anti-reflective film contains the characteristic peak of water absorption within the band. Moisture absorption will greatly affect the performance of anti-reflective film. Therefore, it should be considered to increase the aggregation density of film while filling $F^-$ ion vacancies. Sun discussed the effects of $MgF_2$ prepared at different temperatures on the spectral transmittance and absorption of the deep ultraviolet band produced by the MgO content in the film layer placed in air, which reduced the absorption of the color center [15]. It should be noted, the oxygen can reduce the $F^-$ vacancy. In the paper the effect of oxygen ion source-assisted deposition on the properties of $MgF_2$ is studied. In order to obtain $MgF_2$ thin films with high density and low absorption by $O^{2-}$ assisted deposition. The problem of poor bonding between $MgF_2$ thin films and Si substrates in the experimental process was also analyzed, and the ion source-assisted deposition technology was optimized to improve the adhesion of $MgF_2$ thin films on Si substrates assisted by $O_2$ ion source.

## 2. Experiment

### 2.1. Thin Film Preparation

The films were prepared by electron beam evaporation with $O_2$ ion beam-assisted deposition. The vacuum chamber was evacuated to a base pressure of less than $1.0 \times 10^{-3}$ Pa. With a substrate temperature of 300 °C, thin films were deposited on crystal silicon ($\varphi$20 mm × 1 mm) and float glass ($\varphi$25 mm × 1 mm) with a deposition rate of 0.8nm/s with different ion beams. The substrates were cleaned by ultrasonication. The experiment used a quartz crystal to monitor the physical thickness of the film. The monitor was SQC-310 produced by Inficon Co., Ltd (Inficon, Shanghai, China). To investigate oxygen ion effects on $MgF_2$ film properties, samples were deposited under various oxygen ion currents (Kaufman Ion beam assistant) of 0, 50, and 80 mA at deposition temperature of 300 °C and 0.8 nm/s deposition rate. Sample thicknesses were controlled at 700 nm.

### 2.2. Film Characterization

The properties of $MgF_2$ samples deposited under various oxygen ion currents, such as the crystal frequency, optical transmittance, chemical phase of the film, surface roughness, and adhesion, were analyzed using the techniques listed in Table 1.

**Table 1.** Measurement process.

| Group | Equipment and Accuracy | Purpose | Process |
|---|---|---|---|
| 1 | Inficon golden crystaluency ($\Delta f < 0.004657$ Hz) | Peaking density | 1. Recorded the crystal frequency after coating at pressure of $1.0 \times 10^{-3}$ Pa, recorded the crystal frequency, vented and recorded again. 2. The crystal was taken out and soaked in the ionized water for 36 h to make the crystal oscillator fully absorb water and then the frequency of the crystal oscillator was displayed in the XTC-3100 quartz monitoring system until the frequency stabilized. 3. Thereafter, the crystal was placed in the chamber in which it was pumped to $1.0 \times 10^{-3}$ Pa again at a baking temperature of 300 °C, and the vibration frequency of the crystal piece was recorded. |
| 2 | SHIMADZU UV-3150 ($\Delta T < 0.0002\%$ T) | UV–Visible spectrum | After deposition, the film deposited on BK7 substrate was measured, optichar software (version 9.51) was used to fit the transmittance curve and calculate the optical constant. The Cauchy dispersion module homogeneous layer with a smooth surface was used to fit the optical constant. |
| 3 | Agilent VR610 ($\Delta T < 0.05\%$ T) | Mid-infrared spectrum | After deposition, the film transmittance deposited on Si substrate was measured, and the –OH absorption of different samples were analyzed. |
| 4 | Bruker D8 Advance XRD ($\Delta \theta < \pm 0.001°$) | Chemistry phase of film | X-ray diffraction studies were performed with a Bruker D8 Advance XRD instrument equipped with Cu Ka source, the scan type was "absolute" and scan mode was continuous |
| 5 | Zygo interferometer ($\Delta \lambda < 1/2000\lambda$) | The films' stress | The Zygo interferometer tested the $\Delta$power value then calculated the stress of film. |
| 6 | Adhesive experiments | 3M adhesive tape | Test was done using 3M scotch 610 test tape: Stickiness of $(10 \pm 1)$ N/25mm, and keeping tape pull angle at 90° to the film surface. |

### 2.3. Some Background for Characterizations

#### 2.3.1. Quartz Frequency Measurement of Density

The refractive index of the film will change with the absorption of moisture by the pores within the film. Usually, the packing density or porosity of thin films can be calculated using linear interpolation formula according to the change of refractive index [16]. The refractive index of dense $MgF_2$ film is about 1.38, which is very close to the refractive index of water 1.33. The refractive index change of the film layer after moisture absorption is very small, and it is difficult to calculate the density of the film through the change of refractive index. Therefore, in this study, the density of porous $MgF_2$ was calculated based on the measurement of the mass change of film before and after moisture absorption. A quartz crystal plate is very sensitive to the change of mass. Therefore, the density or porosity of the film layer can be obtained accurately through the change of frequency of quartz crystal plate before and after the absorption of water. The derivation process of the specific calculation is as follows [17].

Its natural frequency is inversely proportional to the thickness t and is proportional to the frequency constant $N$. The relationship is shown in Equation (1)

$$f = N/t \tag{1}$$

$$\Delta f = -\frac{N\Delta t}{t^2} \tag{2}$$

Differentiating Equation (1) with respect to the thickness to get the crystal frequency:

After $MgF_2$ film is deposited, its density is close to that of quartz and the thickness of the coated film during the experiment is much smaller than that of the quartz crystal oscillator, the increase in film thickness can be approximated as the change of the thickness of the quartz crystal.

$$\Delta m = A \bullet \rho_M \bullet \Delta t_M = A \bullet \rho_Q \bullet \Delta t \tag{3}$$

where $\Delta m$ represents the mass change, $A$ is the crystal coating area, and $\rho_M$ and $\rho_Q$ are the film density and the quartz density, respectively. By substituting into Equation (2), the relationship between the change in the quartz crystal frequency and the film thickness can be obtained by Equation (4).

$$\Delta f = -\frac{\rho_M}{\rho_Q} \bullet \frac{f^2}{N} \Delta t_M \tag{4}$$

It can be seen from the formula that the decrease in the quartz crystal frequency is proportional to the thickness and density of the coated film.

Assume that the initial frequency of the crystal is $f_1$, the frequency after water absorption is $f_1^*$, the bulk density of the film is $\rho_s$, and the density is shown in Equation (5),

$$p = \frac{\Delta f_1}{\Delta f_1 + \rho_s \Delta f_1^*} \tag{5}$$

According to Equation (5), the density under various conditions can be easily calculated.

#### 2.3.2. Zygo Interferometry for Measuring Film Stress

Stoney proposed calculating the stress of a thin film by measuring the radius of the deformation curvature of the film and the substrate surface. When the actual thickness of the film is considered to be infinitesimal relative to the substrate, both the film and the substrate can be considered as homogeneous and isotropic materials [18]. The Stoney formulate is shown below:

$$\sigma_f = \left(\frac{E_s}{1-v_s}\right)\frac{t_s^2}{6Rt_f} \tag{6}$$

where, $Es$ and $V_s$ are the elastic modulus and Poisson's ratio of the substrate, respectively. $T_s$ and $t_f$ are the thickness of the substrate and the film, respectively. $R$ is the radius of curvature of the substrate. Based on the interference principle, the Zygo interferometer calculates the curvature change of the measured object by comparing the interference fringe generated by the optical path difference between the measured light and the reference light reflected from the measurement plane [19]. By processing the test results of the Zygo interferometer, the power value indicating the apparent degree of surface deformation can be obtained. The power is showed in Equation (7):

$$power = \frac{D_s{}^2}{8R} \tag{7}$$

where $D_s$ is the diameter of the substrate, and the radius of curvature before and after coating can be expressed by Equation (8),

$$\frac{1}{R_2} - \frac{1}{R_1} = \frac{8}{D_s^2}(power_2 - power_1) \tag{8}$$

Combining Equation (6), the surface stress of the thin film can be expressed by Equation (9),

$$\sigma = \frac{4Es}{3(1-v_s)} \frac{t_s^2}{t_f D_s{}^2} \Delta power \tag{9}$$

## 3. Result and Analysis

### 3.1. Packing Density Character

The variation of crystal oscillator frequency with time after coating is shown in the Figure 1.

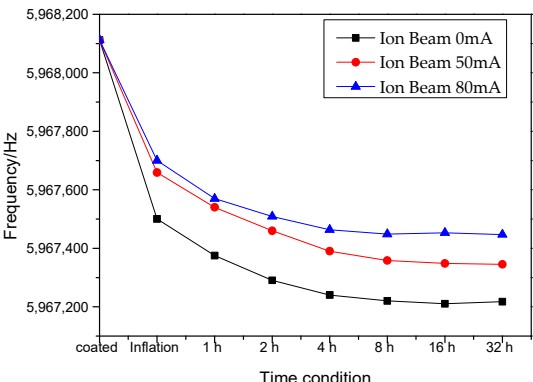

**Figure 1.** Crystal frequency changed with time in different ion beams (the black, red and blue lines represent the frequency with ion beam assistant condition in ion beams 0, 50, and 80 mA, respectively).

As can be seen from the figure, during the venting of the vacuum chamber to the atmospheric state, the frequency of the crystal vibrator is significantly decreased. As the immersion time in the deionized water increases, the frequency of the crystal vibrator decreases continuously, and finally reaches a stable value. The value in the crystal vibrator frequency decreases as the ion beam density increases. According to the formula listed in Section 2.1, the crystal oscillator frequency variation value is used to calculate the packing density of the film. The aggregation densities under different processing conditions are shown in Table 2.

It can be seen from the table that as the oxygen ion beam increases, the density of the film increases. Therefore the oxygen ion-assisted deposition can be an effective technique to reduce the porosity of $MgF_2$ film.

As discussed above, the deposited $MgF_2$ films on quartz crystal were immersed in DI water for a certain time to allow the void of film to be filled by water, and oscillation frequencies of the crystal

were recorded. Then the samples were placed back in a chamber under high vacuum and at high temperature to remove the water trapped in the void. The oscillation frequencies of crystal were also recorded. Table 3 below shows the frequencies of crystal with $MgF_2$ film with and without adsorbed moisture, and the frequency differences.

**Table 2.** The density of the $MgF_2$ films produced at various oxygen ion currents.

| Ion Beam | Density |
|:--------:|:-------:|
| 0 | 0.89 |
| 50 | 0.905 |
| 80 | 0.92 |

**Table 3.** The crystal frequency under various conditions.

| Oxygen Ion Current | 0 mA | 50 mA | 80 mA |
|:-------------------|:----:|:-----:|:-----:|
| Before water immersion (A) | 5,968,113.2 | 5,968,115.3 | 5,968,110.7 |
| After water immersion (B) | 5,967,217.3 | 5,967,345.3 | 5,967,447.3 |
| Under vacuum at 300 °C (C) | 5,967,982.4 | 5,968,003.2 | 5,968,092.6 |
| Frequency differences (C–B) | 761.5 | 657.9 | 645.3 |
| Frequency differences (C–A) | −130.8 | −112.1 | −18.1 |

As the ion beam current increases, the frequency difference between the crystal oscillators before and after water immersion decreases (C–A in Table 4). When the ion beam density is 80 mA, the frequency of the crystal oscillator before (A) and under vacuum after the immersion (C) is very close. It shows that under ion beam condition of 80 mA, the main reason for the change of the crystal vibration frequency is that water vapor penetrates into the pores of the film, and only physical adsorption occurs mainly. Water molecules trapped in the pores with strong bonding are almost negligible. When oxygen ion current is 0 mA, the difference between the crystal oscillators frequency under vacuum at 300 °C (C in Table 4), and the frequency just after the completion of $MgF_2$, is much higher compared to the one under 80 mA oxygen ion current, which means there is a much stronger bonding force between water molecule and $MgF_2$. This indicates that the water vapor is not only physically adsorbed within the deposited film on the crystal oscillator, but also undergoes a chemical reaction to form a strong bond. Through the above tests, it was shown that oxygen ion bombardment introduces modifications of $MgF_2$ film when the $MgF_2$ film was deposited using electron beam evaporation under oxygen ion assistant. The participation of oxygen ion would supplement the crystal defects, improve the crystal structure, and cause the modification of the $MgF_2$ film surface state, including the surface of pores in film (e.g., Mg suspending bond). Oxygen ion also improves the density of the film. From a macroscopic point of view, the refractive index of the film increases, absorption and scattering decrease, and tensile stress decreases.

**Table 4.** Transmittance change.

| Film Condition | Ion Beam 0 mA | Ion Beam 50 mA | Ion Beam 80 mA |
|:--------------:|:-------------:|:--------------:|:--------------:|
| After coated | 3.22% | 1.74% | 0.83% |
| 32 h dipped | 5.65% | 2.64% | 1.52% |
| After vacuum baked | 5.2% | 1.98% | 0.96% |

*3.2. Optical Properties*

Visible/near infrared and mid-infrared transmittance were measured. Visible/near infrared transmittance was used to fit the optical constant and the mid-infrared transmittance was used to analyze the absorption of hydroxyl near 2940 nm.

### 3.2.1. The Visible/Near Infrared Analysis

The Visible-near infrared band film transmittance curve is shown in the Figure 2.

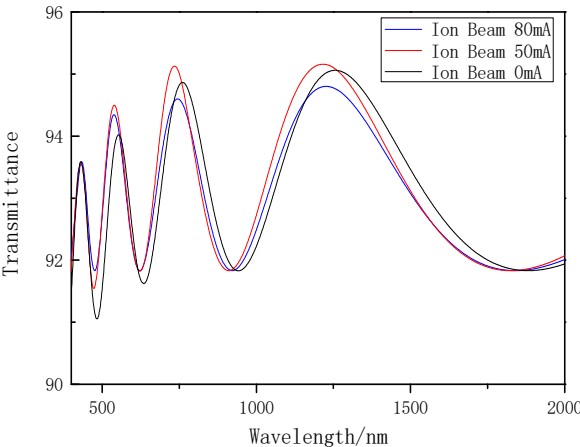

**Figure 2.** The transmittance of different ion beams in the Vis-infrared band (the black, red, blue lines represent transmittance with the ion beam assistant condition in ion beams 0, 50, and 80 mA, respectively).

It can be seen from the figure that there are significant differences in the refractive index and absorption of $MgF_2$ films under different ion beam flow conditions. The optical constant dispersion curve of the film obtained by the full spectrum fitting method is shown in Figure 3.

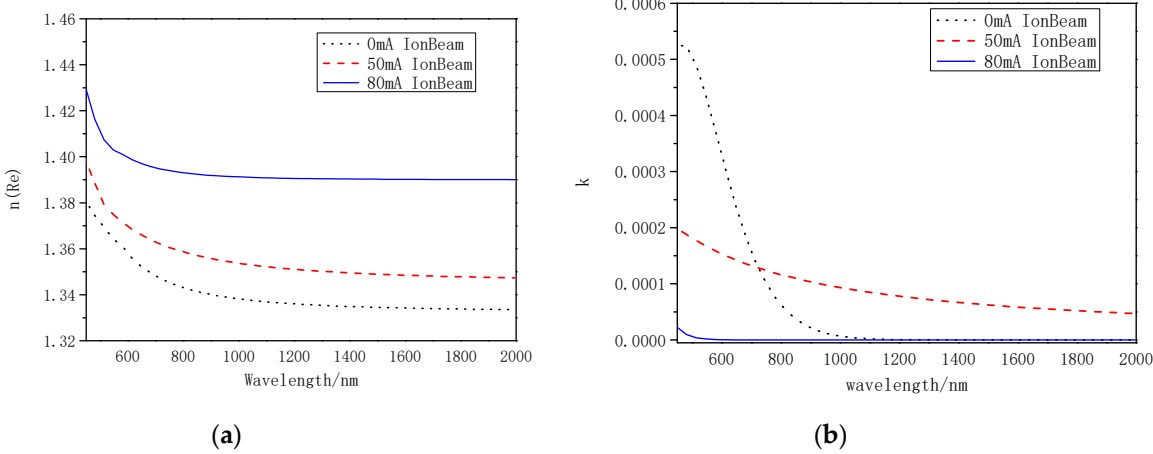

| (a) | (b) |

**Figure 3.** Optical constant dispersion curve (**a**) is the refractive index, (**b**) is the extinction coefficient the black, red and blue lines represent optical constant with the ion beam assistant condition in ion beams 0, 50, and 80 mA, respectively.

It can be seen that the $O_2$ ion beam causes the refractive index to increase. It was because of the presence of MgO compound in $MgF_2$ film. Using the Wiener bounds model, the amount of MgO roughness was estimated as 13.1%.

### 3.2.2. Hydroxyl Vibration Absorption

The infrared transmittance curve of the single crystal Si substrate [20] and the transmittance of the $MgF_2$ film prepared under different ion source conditions are shown in Figure 4.

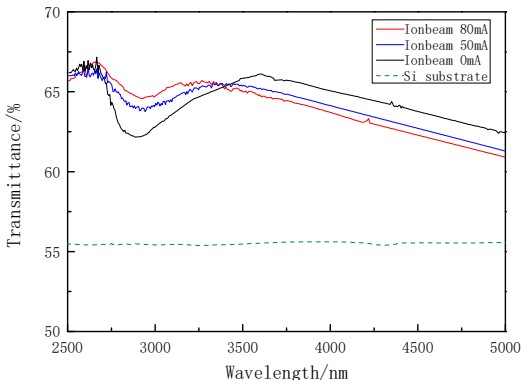

**Figure 4.** The transmittance of MgF$_2$ and substrate (the black, red, blue lines represent mid-infrared transmittance with the ion beam assistant condition in ion beams 0, 50, 80 mA, respectively. The dashed line is Si substrate).

It can be seen that the spectral transmittance of the monocrystalline Si substrate at the hydroxyl vibration position of 2770~3200 nm also fluctuates, indicating that the compounds containing –OH exist on the surface of the Si substrate [21]. The spectral transmittance of MgF$_2$ prepared under different ion source conditions at the band of 2770~3200 nm is significantly decreased, and the characteristic absorption peak of –OH appears at 2940 nm. When the substrate was immersed in deionized water, the spectral transmittance decreases in accordance with the frequency change of the crystal oscillator of Section 3.1. As the immersion time increases, the spectral transmittance decreases continuously and is substantially stable after 32 h. After baking under vacuum for two hours, the infrared spectrum of 2500~5000 nm was retested. The average reduction values of the spectral transmittance before and after baking in the range of 2770~3200 nm are shown in Table 4.

From the table it can be seen that ion beam-assisted deposition can effectively reduce the absorption of MgF$_2$ in the range of 2770 to 3200 nm, and the spectral transmittance decreases the least when the ion beam is 80 mA. After baking under vacuum, the decrease of the spectral transmittance of MgF$_2$ with an ion beam density of 80 mA at 2770 to 3200 nm is basically the same as that before immersion, indicating that the moisture physically adsorbed in the film causes the decrease in the spectral transmittance of the film. While, without using ion bombardment, the spectral transmittance of the film is only changed by 0.4% after baking under vacuum conditions. After baking in vacuum, the film still contains compounds with hydroxyl groups.

This indicates that the decrease of absorption of MgF$_2$ thin film prepared by oxygen ion source-assisted deposition near 2940 nm is not only due to the increase of concentration density of the film layer, but also due to the recombination of oxygen ion and F$^-$ vacancy caused by Mg suspension bond during deposition, preventing the suspension bond from forming other compounds with hydroxyl in water vapor.

*3.3. XRD*

According to the result of spectrum test, the oxygen ion and Mg suspending bond fill the film defects, reduce the colour center absorption, and reduce the hydroxyl absorption of film by filling the F$^-$ vacancy with O particles. To analyze the phase change of film, an XRD measurement was taken, the result are shown in Figure 5.

The XRD results show that MgF$_2$ film exhibits a polycrystalline state, and the film has a remarkable crystal orientation, with the maximum intensity of the diffraction peak in the <110> direction. It can be seen that the location of the diffraction peak without assisted deposition by oxygen ion source is consistent with that of the standard PDF (powder diffraction file) card, as the oxygen ion beam flow increases, the diffraction peak positions in each direction shifts to the right, the intensity of the diffraction peak decreases, and the diffraction peak gradually widens as the ion beam flow increases.

When the ion beam was increased to 80 mA, the $MgF_2$ diffraction peak in the <220> direction and significantly broadened. The diffraction peak became inconspicuous. In addition, the characteristic diffraction peak of MgO appears at positions where the diffraction angle (2θ) values are 36.703° and 62.219°, indicating that under the beam condition the O ion compounds with Mg in the film to form a MgO crystal.

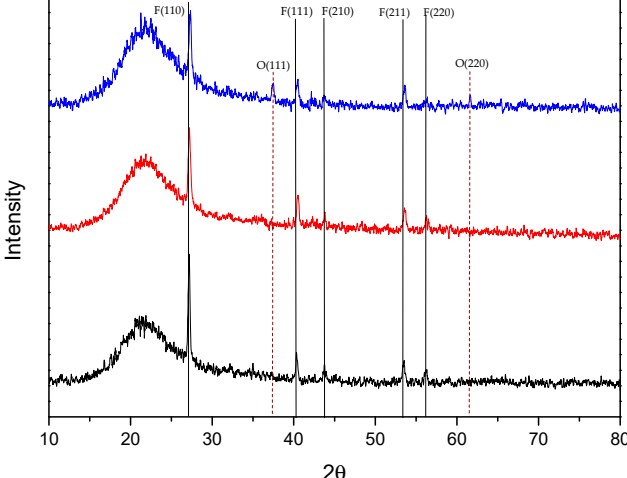

**Figure 5.** XRD measurement of different ion beams (the blue, red, and black represent ion beams 80, 50, and 0 mA, respectively; the vertical lines represent the theoretical positions of the peaks for MgF2 and MgO; F represents $MgF_2$, O represents MgO).

The comparison parameters of the crystal plane spacing of $MgF_2$ thin film and PDF standard card are as per Table 5.

**Table 5.** The crystal plane spacing of sample and PDF standard value.

| Spacing (d) <hkl> | Literature ICCD41-1443 | Sample 1 Ion Beam 0 mA | Sample 2 Ion Beam 50 mA | Sample 3 Ion Beam 80 mA |
|---|---|---|---|---|
| <110> | 3.2670 | 3.2668 | 3.2683 | 3.2686 |
| <111> | 2.2309 | 2.2299 | 2.2312 | 2.2315 |
| <210> | 2.0672 | 2.0670 | 2.0673 | 2.0674 |
| <211> | 1.7112 | 1.7113 | 1.7120 | 1.7124 |
| <220> | 1.6335 | 1.6338 | 1.6343 | 1.6345 |

The lattice constants of the $MgF_2$ film and the PDF reference card prepared under different process conditions were calculated by Jade software and are shown in Table 6.

**Table 6.** The crystal character calculates by MDI Jade.

| Lattice | Literature ICCD41-1443 | Sample 1 Ion Beam 0 mA | Sample 2 Ion Beam 50 mA | Sample 3 Ion Beam 80 mA |
|---|---|---|---|---|
| a (nm) | 0.46200 | 0.46220 | 0.4295 | 0.4635 |
| C (nm) | 0.30509 | 0.30622 | 0.3041 | 0.3032 |

As can be seen, with the increase of oxygen ion beam, the crystal axis cell becomes smaller and the spacing between the crystal planes decreases. The oxygen ion source-assisted deposition indicates that the crystal cells gather more closely during the growth of the film.

### 3.4. Adhesion

The tape peeling method was used for adhesion test, where tape is attached to the sample and then peeled off. The adhesion is then judged on the extent of the film delamination. The measurement process is simple and has good repeatability. The test was done using 3M scotch 610 test tape: Stickiness of $(10 \pm 1)$ N/25mm, and keeping tape pull angle at 90° to the film surface [22].

The film without $O_2$ ion assistant deposition can endure being peeled off 20 times. When the $O_2$ ion beam was 50 mA, the film delaminated after being peeled off 15 times. When the ion beam increased to 80 mA the film delaminates after being peeled off 12 times. Indicating the film adhesion decreases with increasing ion beam. The adhesion of the film is mainly affected by stress and inter-material adsorption [23]. Thus, the stress was discussed as follows.

The changes in curvature of the $MgF_2$ film before and after deposition obtained by a Zygo interferometer are shown in Table 7. According to the formula in Section 2.2, the stress of the film is 986, 436, and 357 MPa, respectively. It can be seen that the tensile stress of the film decreases as the ion beam density increases.

**Table 7.** Curvature of substrate before and after coating.

| Ion Current | Power before Coated | Power after Coated |
|---|---|---|
| Ion Beam 0 mA | 0.091 | 0.866 |
| Ion Beam 50 mA | 0.091 | 0.425 |
| Ion Beam 80 mA | 0.091 | 0.365 |

The stress test results show that when the ion beam is 80 mA, the tensile stress of the film is at the minimum, and the stress of the film should not be the main cause for poor adhesion. It can be seen from the infrared spectrum test results that at 2770~3200 nm the Si substrate has obvious absorption, indicating that the surface of the Si substrate contained the OH root compound. When the $MgF_2$ film was modified without using oxygen ions, there were obvious $F^-$ vacancies in the film layer. The free $Mg^{2+}$ ions were combined with the hydroxyl of the Si substrate. The film and the substrate were combined by the bonding force. The $MgF_2$ film substrate, in which the F ionic vacancies were filled by the O particles, was combined by van der Waals force. Therefore, the film–substrate adhesion is much smaller than the bonding force [24,25]. In order to obtain a $MgF_2$ film with good adhesion to the substrate, and effectively reducing the absorption of water molecules, ion beam-assisted deposition was not used at the beginning of up to 50 nm thickness of deposition, and a film (transition layer film) having $F^-$ vacancies was obtained. The –OH compounds formed by bonding with the Si substrate made the film adhere well to the substrate. Next, O ion bombardment by increasing the beam density to 80 mA was used to fill the $F^-$ ion vacancies in the film during deposition. The comparison of 3M adhesive experiments before and after adjusting the ion source is shown in Figure 6.

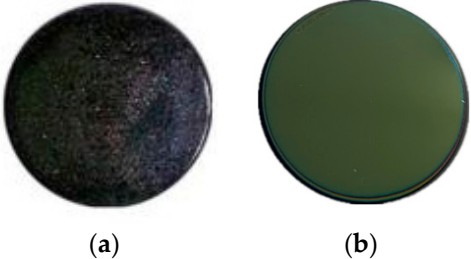

(**a**)         (**b**)

**Figure 6.** Sample condition after tape pull test: (**a**) Without transition layer which was produced by electron beam vapor deposition and $O_2$ IAD; (**b**) the first transition 50 nm deposited using only electron beam vapor deposition, then deposition completed with IAD.

In addition, the comparison of transmittance before and after adjusting the ion source is shown in Figure 7.

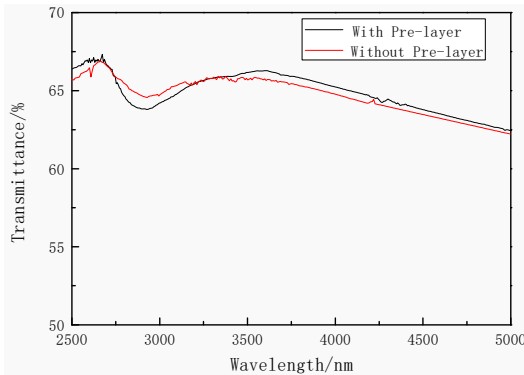

**Figure 7.** The transmittance of MgF$_2$ (the black and red line represent mid-infrared transmittance with 50 nm pre-layer and without pre-layer, respectively.)

## 4. Conclusions

The influence of oxygen ion source on MgF$_2$ film's visible near-infrared band and mid-wave infrared absorption was studied. As the oxygen ion beam flow increases, the film aggregation density increases. Due to the oxygen ions filling into the colour center defects, generated by F$^-$ vacancies, an MgO compound is formed in the film, so that the absorption value of the film in the visible range is reduced, and the refractive index is increased. In the infrared light portion, as the oxygen ion beam increases, the F$^-$ ionic vacancies are filled with O, preventing the Mg$^{2+}$ ions from recombining with the –OH in the air, and reducing the absorption of the film in the infrared. However, the stress and adhesion test results show that as the oxygen ion beam flow increases, the bonding force between the film and the substrate changes from chemical bonding force to van der Waals force, and the tensile stress of the MgF$_2$ film decreases, which lead to a bad adhesion for MgF$_2$ film on Si substrate. A solution of two steps of deposition was proposed to solve the problem of poor adhesion: At the early stage of deposition of the MgF$_2$ film, a very thin adhesion layer without ion beam-assisted was formed and allowed further MgF$_2$ film to bond to the substrate with good adhesion, then the O$^+$ ion current for further MgF$_2$ deposition was increased to allow O to fill the F anion vacancies efficiently, and reduce the absorption. The experimental results demonstrate that MgF$_2$ film with low absorption, high stability, and good adhesion was achieved. The effect of the thickness of the thin adhesive MgF$_2$ layer deposited without ion beam assisted requires further investigation.

**Author Contributions:** Conceptualization, G.Z. and X.F.; methodology, G.Z.; software, G.Z.; validation, S.S., K.G. and G.Z.; formal analysis, S.S.; investigation, J.Z.; resources, J.Z.; data curation, S.S.; writing—original draft preparation, G.Z.; writing—review and editing, S.S.; visualization, K.G.; supervision, X.F.; project administration, X.F.

**Funding:** This research received no external funding.

**Conflicts of Interest:** The authors declare no conflicts of interest.

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
