# Peer review of "Influences of Oxygen Ion Beam on the Properties of Magnesium Fluoride Thin Film Deposited Using Electron Beam Evaporation Deposition"

_coatings, doi:10.3390/coatings9120834_

Round 1

Reviewer 1 Report

This paper describes properties of magnesium fluoride coatings prepared by e-beam evaporation assisted by oxygen ion bombardment. The authors find that oxygen ion bombardment can affect the layer properties and suggest that the formation of MgO compounds is responsible for absorption reduction. The results are interesting, and the paper content falls into the scope of „Coatings“. However, some more information should be included into the manuscript to enable the reader to follow the authors argumentation. My proposals are:

Line 37: include a reference Lines 41 – 46: This problem is not so specific to MgF2. Ion assisted deposition of several fluorides (MgF2, AlF3, YbF3, LaF3,…) has been studied and reported in earlier work. In order to place the study into the proper context, I would recommend including a short discussion of relevant earlier work (e.g. Targove et al 1988, Kennedy et al 1989, Bischoff et al 2011) on fluorides into the introduction. 2.1: I could not find information about the ion beam source used. Please include. 1: Has the transmittance really been fitted by OptiLayer? I guess it has been done by OptiChar. (1) – Was unreadable in my pdf version… (2) and (6): different symbols for the film thickness used Line 174: „spectrum fitting method“ – this information is insufficient. Please indicate at least the layer model (homogeneous or inhomogeneous, smooth or rough surfaces,…) and the dispersion model used. 4 and 5 may be merged into one figure. Lines 277 and 278: When looking at Fig. 3a, the refractive index indicated in blue is really remarkably high, and that could be a result of the presence of MgO compounds in the MgF2. If so, then the amount of MgO could be easily estimated in terms of a mixing model or simply from the Wiener bounds. Such an estimation should be provided in order to discuss the physical consistency of that assumption.

Author Response

Dear reviewer:

    Thank you so much for careful and thorough reading of this manuscript and for the thoughtful comments and constructive suggesting.

please see the attachment of the response.

Reviewer 2 Report

COMMENTS TO AUTHOR:

In the work entitled “Influences of oxygen ion beam on the properties of magnesium fluoride thin film deposited using electron beam evaporation deposition”, G.Zhang et al. use electron beam  evaporation with oxygen ion assistant for the growth of magnesium fluoride thin films. They study the optical properties of the grown material as a function of the oxygen ion dose showing that this parameter influences also the crystalline phase of the films.

The topic can be of potential interest for the audience of the journal, however, I have some concerns on the scientific soundness of the following points:

1.

Paragraph 2.4.2, please provide correct and adequate references for the Zygo interferometry and Stoney formula.

2.

the optical constants derived in paragraph 3.2.1 are very different for the three samples (Figure 3), however the transmittance spectra of Figure 2 don’t show such significant differences. Why so? Which kind of fitting algorithm has been used for the calculation? What are the free parameters of the fitting? How did the authors assess them?

3.

The average optical transmittance of the silicon substrate reported in Figure 4 is 56%. What is the reference of this spectrum? Why the transmittance of the sample increases after the deposition of the MgF films (Figure 5)?

4.

Consider improving the quality of the plot of Figure 1. What does it mean “Non IAD” in the legend?

The authors claim that the introduction of a 50 nm thick transition layer improves the adhesion of the film onto the substrate. How do the optical properties of the film change with this extra layer?

In my opinion, the manuscript is concise and well-written and can be considered for publication after revision and comment to the raised points.

Author Response

Dear reviewer:

Thank you so much for careful and thorough reading of this manuscript and for the thoughtful comments and constructive suggesting.

Please see the attachment for the response.

Reviewer 3 Report

The manuscript by Zhang et al. describes the study of magnesium fluoride thin films deposited by EBE and the influence of oxygen on its properties. Authors have used the broad spectrum of experimental techniques to analyze structural, optical and mechanical properties of produced layers. In my opinion, the manuscript could be accepted after some minor revisions. I would suggest to add more novel references in the introduction and to point the novelty of the proposed research. Besides, please add a confidential interval/error to all physical values. 

Author Response

(The authors gave the same response as above.)

Round 2

Reviewer 1 Report

The topic of the paper is of current interest. In the revised version, the authors have considered all of my remarks, so that - in my view - a sufficient amount of information is given to the reader now to make the approach of the authors understandable.

Optional: A reference to the Wiener bounds is missing, and stylistic improvements (particularly with respect to the new passages) would enhance the value of the paper.

Reviewer 2 Report

The authors have address most of my comments/suggestions in their revised version. Therefore, I recommend the publication of the manuscript in the present form.